# Impact of Optimized Ku–DNA Binding Inhibitors on the Cellular and In Vivo DNA Damage Response

**DOI:** 10.3390/cancers16193286

**Published:** 2024-09-26

**Authors:** Pamela L. Mendoza-Munoz, Narva Deshwar Kushwaha, Dineshsinha Chauhan, Karim Ben Ali Gacem, Joy E. Garrett, Joseph R. Dynlacht, Jean-Baptiste Charbonnier, Navnath S. Gavande, John J. Turchi

**Affiliations:** 1Department of Medicine, Indiana University School of Medicine, Indianapolis, IN 46202, USA; 2Department of Pharmaceutical Sciences, Eugene Applebaum College of Pharmacy and Health Sciences, Wayne State University, Detroit, MI 48201, USA; 3Institute for Integrative Biology of the Cell (I2BC), Institute Joliot, CEA, CNRS, Université Paris-Sud, 91198 Gif-sur-Yvette Cedex, France; 4Structure-Design-Informatics, Sanofi R&D, 94400 Vitry sur Seine, France; 5Department of Radiation Oncology, Indiana University School of Medicine, Indianapolis, IN 46202, USA; 6Molecular Therapeutics Program, Barbara Ann Karmanos Cancer Institute, Wayne State University, Detroit, MI 48201, USA; 7NERx Biosciences, Indianapolis, IN 46202, USA

**Keywords:** DNA-PK, non-homologous end joining, double-strand break repair, Ku-DBis, small-molecule inhibitors, non-small cell lung cancer

## Abstract

**Simple Summary:**

DNA-dependent protein kinase (DNA-PK) is a key player in repairing DNA damage. Ku proteins detect DNA damage and activate DNA-PK. Blocking DNA-PK can make cancer treatments such as radiation more effective. We have developed Ku–DNA binding inhibitors (Ku-DBis) that stop DNA-PK activation, making cancer cells more sensitive to radiation. We recently discovered new Ku-DBis that enter cells better than predecessor compounds, allowing cellular and in vivo analyses. Some non-small cell lung cancer (NSCLC) cells, especially those lacking ATM, were very sensitive to these inhibitors and radiation. However, cells lacking BRCA1 were resistant, which reduced the effectiveness of radiation. In animal studies of NSCLC, Ku-DBi treatment inhibited DNA-PK and enhanced a radiation-dependent decrease in tumor cell proliferation. This is the first time a Ku-targeted inhibitor has been shown to work in living organisms, suggesting their potential application in cancer treatment.

**Abstract:**

**Background**: DNA-dependent protein kinase (DNA-PK) is a validated cancer therapeutic target involved in DNA damage response (DDR) and non-homologous end-joining (NHEJ) repair of DNA double-strand breaks (DSBs). Ku serves as a sensor of DSBs by binding to DNA ends and activating DNA-PK. Inhibition of DNA-PK is a common strategy to block DSB repair and improve efficacy of ionizing radiation (IR) therapy and radiomimetic drug therapies. We have previously developed Ku–DNA binding inhibitors (Ku-DBis) that block in vitro and cellular NHEJ activity, abrogate DNA-PK autophosphorylation, and potentiate cellular sensitivity to IR. **Results and Conclusions**: Here we report the discovery of oxindole Ku-DBis with improved cellular uptake and retained potent Ku-inhibitory activity. Variable monotherapy activity was observed in a panel of non-small cell lung cancer (NSCLC) cell lines, with ATM-null cells being the most sensitive and showing synergy with IR. BRCA1-deficient cells were resistant to single-agent treatment and antagonistic when combined with DSB-generating therapies. In vivo studies in an NSCLC xenograft model demonstrated that the Ku-DBi treatment blocked IR-dependent DNA-PKcs autophosphorylation, modulated DDR, and reduced tumor cell proliferation. This represents the first in vivo demonstration of a Ku-targeted DNA-binding inhibitor impacting IR response and highlights the potential therapeutic utility of Ku-DBis for cancer treatment.

## 1. Introduction

The DNA damage response (DDR) is a collection of signaling pathways initiated by independent DNA-binding proteins that sense DNA damage and specific DNA structures including DNA double-strand breaks (DSBs). These sensors, namely MRE11, RPA, and Ku then activate the phosphatidylinositol-3-kinase-related kinases (PIKKs) ATM, ATR, and DNA-PK, respectively, to regulate cell cycle, cell death, DNA replication, and DNA repair [1]. Targeting the DDR is a promising therapeutic strategy, as many therapeutic modalities induce cancer cell death through generation of DNA damage.

DSBs are induced by ionizing radiation (IR), radiomimetic agents, and other chemotherapeutics that directly damage DNA. DSBs can also be induced during repair of DNA damage. A variety of pathways can repair DNA DSBs; however, repair-proficient cells will mostly employ the highly efficient and cell cycle-independent, error-prone non-homologous end-joining (NHEJ) pathway [2]. NHEJ is initiated by binding of the Ku heterodimer to DNA termini, creating a platform for binding and subsequent activation of DNA-PKcs. The importance of DNA-PK in IR response has led to development of several small-molecule DNA-PK inhibitors that target kinase active sites, and these inhibitors are being pursued both pre-clinically and clinically [3]. As an alternative to ATP-mimetic kinase inhibitors, we first reported the discovery of Ku–DNA binding inhibitors (Ku-DBis), demonstrating in vitro inhibition of Ku binding and DNA-PK activation [4]. Through initial optimization of Ku-DBi, we identified a series of compounds with potent in vitro inhibition of DNA-PK- and NHEJ-catalyzed DSB repair [5]. Although limited cellular uptake of these Ku-DBis necessitated use of serum-free media for cellular Ku-DBi treatment, we were able to demonstrate on-target cellular activity and sensitization to IR and other DSB-inducing treatments [5,6].

In this report, we performed further chemical optimization, leading to the discovery of a series of oxindole derivatives based on our earlier Ku-DBi core structure. These novel Ku-DBis exhibited improved cellular uptake and retained potent Ku inhibitory activity. Interestingly, HR-deficient triple-negative breast cancer (TNBC) cells were more resistant to Ku-DBis and displayed antagonism with DSB-inducing agents. On the other hand, the results obtained in NSCLC models were markedly different, with the results in ATM-null DSB repair-deficient lines displaying high sensitivity and synergy with IR. In addition, we demonstrated in vivo activity in a NSCLC xenograft model where IR-induced DNA-PK autophosphorylation was abrogated by pre-treatment with the optimized Ku-DBi. Furthermore, a decreased expression in the proliferation marker Ki-67 was observed in the cells treated with Ku-DBi in combination with IR, as assessed by both Western blotting and immunohistochemistry (IHC). An increase in γ-H2AX was also observed in combination-treated tumors, consistent with ATM activation, as we have previously described in vitro [6]. These data demonstrate the first in vivo assessment of an optimized novel Ku-DBi and pave the way for further development of this anticancer therapeutic strategy in combination with radiotherapy.

## 2. Materials and Methods

### 2.1. Chemistry and Drug Reconstitution

The synthetic schemes, procedures, and characterization of the novel Ku-DBis are provided in the Appendix A. Ku-DBis (10 mM) and NU-7441 (5 mM) (Cat: 3712, TOCRIS, Minneapolis, MN, USA) powder were reconstituted in dimethyl sulfoxide (DMSO) and stored at room temperature and protected from light. Bleomycin stock (2.5 mM) (bleomycin sulfate, cell grade; Cat: J67560-S, Alfa Aesar, Haverhill, MA, USA) was prepared in sterile water and stored at −20 °C.

### 2.2. Protein Purification and Preparation

Purification of Ku 70/80 heterodimer and DNA-PKcs was prepared from baculovirus-infected Sf9 cells and HeLa or HEK-293 cells, respectively, as previously described [7].

For biophysical analyses, full-length Ku heterodimer was expressed in and purified from Sf21 insect cells using a multiBac expression system as previously described [8,9]. Protein labeling was achieved following buffer exchange using an Amicon 50 kDa cutoff concentrator into PBS (10 mM phosphate buffer, pH 7.4, 2.7 mM KCl, 137 mM NaCl) supplemented with 0.005% Tween-20.

### 2.3. Biophysical Analysis

#### 2.3.1. Microscale Thermophoresis (MST)

Ku heterodimer was labeled using the Monolith NT™ Protein Labeling Kit RED-NHS (2nd generation amine reactive, NanoTemper Technologies GmbH, MO-L011, München, Germany), following the manufacturer’s protocol. **Ku-DBi 3392** powder was resuspended in 100% DMSO, and the concentration was adjusted to 50 mM after UV absorbance measurement (absorbance at 417 nm with an extension coefficient equal to 15.2 mM^−1^ cm^−1^). The ligand stock solution was sonicated in an ultrasonic bath before preparations of the dilution series. The MST and Differential Scanning Fluorometry (DSF) assays contain 5% DMSO to fit the DMSO concentration at the highest ligand concentration (200 µM). The labeled protein (5 nM) was incubated with ligand 3392 ranging from 98 nM to 200 µM in a 12-point 1:1 dilution series. Protein–ligand solutions were incubated for 30 min at room temperature before MST measurements, which were performed in triplicate. Proteins were transferred to capillaries (Monolith NT. Automated Premium Capillary Chips), and binding was analyzed with a Monolith NT. Automated pico-RED device using MO. Control Software with nano-red excitation, LED light adjusted to 20% excitation power, and infrared laser (MST power) set to medium. The dissociation constants (*K*_D_) were determined with MO. Binding Affinity software ( NanoTemper Technologies GmbH, v1.6) using the single-site fit binding mode (referred to as *K*_D_ binding mode in the software).

#### 2.3.2. nanoDifferential Scanning Fluorometry (nanoDSF)

The thermostability and aggregation propensity of Ku was assessed using a Prometheus Panta nanoDSF instrument (NanoTemper Technologies GmbH, München, Germany). High-sensitivity capillaries were filled with Ku (10 μL each, experiments in triplicate) at a concentration of 5 µM into the instrument. We titrated against ligand **3392** covering a concentration ranging from 0 to 30 µM with measurements collected at a 1 °C-per-minute scan rate from 25 to 95 °C. Thermal unfolding was assessed by the increase in intrinsic tryptophan fluorescence detected at 350 and 330 nm following excitation at 280 nm. Inflection points in the thermal denaturation curves were identified using the first derivatives of the ratio r = fluorescence (350 nm)/fluorescence (330 nm). We measured dynamic light scattering (DLS) and turbidity throughout the heating ramp using back reflection to characterize protein aggregation. We determined aggregation temperatures using turbidity variations and increases in particle sizes.

### 2.4. Biochemical Activity Assays

The inhibitory activity of Ku-DBis on Ku-DNA binding was evaluated in vitro using Electrophoretic mobility shift assays (EMSA) with a [^32^P]-labeled 30-bp duplex DNA and purified Ku 70/80. Kinase assays were performed to measure the DNA-dependent transfer of [^32^P] from γ-[^32^P] ATP to a synthetic p53-based peptide substrate, both as previously described [7]. Both EMSA and DNA-PK inhibitor titration assays were performed in triplicate. Data were fit to standard binding curves using GraphPad Prism to determine IC_50_ values.

### 2.5. Cell Lines and Cell Culture

Human NSCLC cell lines H460 (HTB-177™), A549 (CCL-185™), H1299 (CRL-5803™), and H23 (CRL-5800™) and triple negative breast cancer (TNBC) cell lines MDA-MB-436 (HTB-130™) and MDA-MB-468 (HTB-132™) were obtained from ATCC^®^.

H460, H1299, and H23 cells were cultured in RPMI-1640 medium (RPMI-1640 with L-glutamine; Cat:10-040-CV, Corning, Corning, NY, USA). A549 cells were cultured in F-12K medium (Ham’s F-12K nutrient mixture, Kaighn’s modified, with L-glutamine; Cat: 10-025-CV, Corning). MDA-MB-436 and MDA-MB-468 cells were cultured in DMEM/F12 mixture (1X Dulbecco’s modification of Eagle’s medium with 4.5 g/L glucose, L-glutamine and sodium pyruvate. Cat: 10-013-CV, Corning/Ham’s F-12, 1X, modified with L-glutamine. Cat: 10-080-CV, Corning). Cell number and seeding details for each experiment are provided in the corresponding assay descriptions. All cell lines were obtained from American Type Culture Collection (ATCC, Manassas, VA, USA).

All media were supplemented with 10% fetal bovine serum and 1X penicillin–streptomycin [Penicillin Streptomycin solution, 100×; penicillin (10,000 IU/mL) and streptomycin (10,000 μg/mL) Cat: 30-002-CI, Corning].

For treatments, the vehicle controls correspond to 1% DMSO. Ku-DBi treatments were prepared in supplemented media at a 1% DMSO final concentration.

### 2.6. Determination of Cellular Drug Uptake

Cellular uptake was evaluated in two NSCLC cell lines, H460 and A549. Cells were seeded into 6-well plates at a density of 1 × 10^6^ cells per well and cultured as monolayers at 37 °C and 5% CO_2_ for 18–24 h. Ku-DBis were added to the media at a concentration of 10 µM and incubation continued for the times indicated. Media were removed, cells were washed 3 times with PBS, 1 mL of methanol was added to each well, and cells were agitated overnight at 4 °C. The methanol was collected, and wells were washed with an additional 1 mL of methanol that was then pooled with the original. The methanol extracts were then dried under vacuum, resuspended in methanol, and analyzed by HPLC using a 250 × 4.6 and 5 μm C-18 reverse phase column with 1% TFA mobile phase in acetonitrile (ACN) gradient elution. Absorption was monitored at 425 nm, HPLC peaks were quantified using a standard curve, and picomoles of compound per million cells was calculated.

### 2.7. Cell Viability Assay Assessment

Cell metabolism/viability was assessed using a mitochondrial metabolism assay (CCK-8) kit (Cell Counting Kit-8, Cat: CK04, Dojindo Laboratories, Rockville, MD, USA) as previously described [6]. Briefly, cells were seeded at a density of 3 × 10^3^ cells per well in 96-well plates and allowed to adhere and stretch for 18–24 h prior to treatment. Cells were pre-treated with Ku-DBi for 24 h before bleomycin treatments. Subsequently, cells were incubated for an additional 48 h at 37 °C after which CCK-8 assay was performed. The absorbance of the formed formazan product in each well was measured at 450 nm and compared to vehicle-treated controls to determine percent cell viability. The results represent the average and SEM of triplicate determinations.

### 2.8. Cell Irradiation and Clonogenic Survival Assays

NSCLC cells were seeded at 2 × 10^5^ cells per well into 24-well plates and grown as monolayers at 37 °C and 5% CO_2_ in media one day before Ku-DBi or vehicle treatments. After 2 h, cells were placed on ice for 10 min and kept on ice during irradiation with either 2 or 5 Gy of 160 kVp X-rays using a Precision X-ray machine (North Branford, CT, USA) at a dose rate of 0.687 Gy/min. Radiation dosimetry measurements were performed using a Farmer-type ionization chamber (PTW Model N30013, Freiburg, Germany) in conjunction with a Keithley electrometer (Model K602, Cleveland, OH, USA). After irradiation, cells were incubated for 1 h at 37 °C, trypsinized, replated in 100 mm dishes, and incubated at 37 °C in 5% CO_2_. After 11 days, cells were carefully washed in PBS before fixing and staining in 0.5% crystal violet (Cat: C581-100, Fisher Scientific, Hampton, NH, USA) with 6% glutaraldehyde (Glutaraldehyde solution, 25%. Cat: O2957-1, Fisher Scientific) in PBS for 1 h at RT. The staining solution was removed, and cells were washed with water and air-dried. For clonogenic survival analysis, colonies containing more than 50 cells were manually counted and normalized to vehicle control conditions to determine surviving fractions.

### 2.9. Cancer Cell Line-Derived Xenograft (CDX) Model Studies

The in vivo studies were conducted as approved by the Institutional Animal Care and Use Committee at Indiana University School of Medicine. A549 cells (~2.5 × 10^6^) in 50% Matrigel were injected into the hind flanks of 8–10-week-old NOD.Cg-Rag1tm1Mom Il2rgtm1Wjl/SzJ (NRG) mice (IVT, In vivo Therapeutics Core, Indiana University Melvin and Bren Simon Comprehensive Cancer Center, Indianapolis, IN, USA) as previously described [10]. Tumor volumes were monitored by electronic caliper measurement. Mice were randomized into groups of 3 or 4 using a random group generator; 6 different scenarios were generated, and the scenario with the most similar tumor volume averages across groups was selected. Treatments were initiated when tumors were ~600 mm^3^. Tumors were administered Ku-DBi (30 µL of 10 mM) or vehicle (30 µL of DMSO) via intratumoral injection 4 h before 5 Gy irradiation of the tumor with a single dose of X-rays (250 kVp; dose rate = 1.4 Gy/min; 2 cm × 2 cm field) under isoflurane anesthesia. Mice that did not receive the irradiation treatment were placed under isoflurane anesthesia for the same length of time as the irradiated mice (~8 min). Mice were sacrificed 2 h post-irradiation (or 2 h post-sham irradiation); tumors were excised and processed for immunohistochemistry (IHC) assays. Cell-free protein extracts were prepared to assess DNA-PKcs autophosphorylation and γ-H2AX levels.

### 2.10. Protein Extraction and Western Blotting

Cell cultures were kept on ice for 15 min before protein extraction. Media was removed, and cells were washed in cold PBS and lysed in RIPA buffer (50 mM Tris, pH 8.0, 150 mM sodium chloride, 1% NP-40, 0.5% sodium deoxycholate, 0.1% SDS, 1 mM EDTA). A protease- and phosphatase-inhibitors cocktail [Halt™ Protease and phosphatase inhibitor, single-use cocktail (Thermo Fisher Scientific, Waltham, MA, USA)] was added to lysis buffer before use. Bath sonication on ice was used for disrupting cellular membranes and to release the cells contents. Lysates were centrifuged at 4 °C for 20 min at 14,000 rpm, and the supernatants were collected. Pellets were saved and processed for γ-H2AX immunodetection from nuclear extracts.

### 2.11. Preparation of Nuclear Extracts from Cellular Lysates or Tissue

Nuclear extracts for γ-H2AX immunodetection were prepared following a protocol modified from Abmayr et al. [11]. Briefly, cell pellets were lysed in 200–300 µL nuclear extraction buffer (50 mM Tris, pH 8.0, 300 mM sodium chloride, 1% NP-40, 0.5% sodium deoxycholate, 0.1% SDS, 1 mM EDTA, 10% glycerol). Protease and phosphatase inhibitors were added before use, and DNA in the samples was sheared using a 21-gauge needle, vortexed every few minutes for 10 min, and sonicated on ice using a Qsonica Sonicator (Q125 Sonicator, Newtown, CT, USA) for 16 s on 0.2 s on/off pulse cycles at 20% amplitude. Samples were centrifuged at 14,000 rpm at 4 °C for 20 min, and supernatant containing the nuclear fraction was collected. Protein content was quantified using a BCA protein assay kit (Pierce™ BCA protein assay kit; Cat: 23227, Thermo Scientific, Waltham, MA, USA).

### 2.12. Electrophoresis and Western Blotting

Protein lysates, ranging from 20 to 30 μg, were separated by SDS-PAGE (4–20% Mini Protean TGX Gels, BioRad, Hercules, CA, USA) and transferred onto PVDF membranes (Bio-Rad) through wet transfer. Membranes blocked in 0.5% Tween and 3% BSA in 1× TBS for 1 h were probed with the following primary antibodies: Phospho-DNA PKcs (S2056) (Cat: ab124918, Abcam, Cambridge, UK), total DNA-PKcs (Cat: sc-5282, Santa Cruz Biotechnology, Dallas, TX, USA), and γ-H2AX pS139 (Cat: 613401, BioLegend, San Diego, CA, USA). Membranes blocked in 0.5% Tween and 5% non-fat dried milk in 1× TBS for 1 h were probed with the following primary antibodies: Ki-67 (Cat: ab16667, Abcam) and GAPDH (Cat: MA-15738, Thermo Fisher, Waltham, MA, USA). All primary antibodies were prepared with 3% BSA and 0.02% NaN_3_ in TBS with 0.5% Tween. Goat anti-rabbit or anti-mouse IgG (H + L)–HRP conjugates (Cat: 170-6515 or 170-6516, Bio-Rad) were used as secondary antibodies to detect these proteins. Imaging, densitometric analysis, and quantification of protein expression were performed as previously described [5,6].

### 2.13. Tissue Sections and Immunohistochemistry (IHC) Staining

Excised tumors were fixed in 4% paraformaldehyde (PFA) solution in PBS for 24 h at 4 °C. After fixation, tumors were transferred to 70% ethanol and paraffin embedding. Sectioning was completed by the Indiana University Histology Core at the IU School of Medicine. Representative 3 μm-thick sections for each treatment were stained with H and E (HE; Harris hematoxylin, regressive method) for histologic examination or were processed for Ki-67 IHC immunodetection. Briefly, paraffin-embedded sections were deparaffinized in a xylene series (4 × 3 min) and rehydrated in a series of graded alcohol dilutions (100%, 95%, 70%, and ddH_2_O, 2 × 2 min each). Ki-67 antigens were retrieved using a low-pH antigen-retrieval solution (10 mM sodium citrate, pH 6.0, Cat: 00-4955-58, Thermo Fisher Scientific) and subsequently cooled on ice for 15 min. Hydrogen peroxide 3% in water reagent was used to block endogenous peroxidase activity (Hydrogen Peroxide 30% *v*/*v*, Cat: H1009 7722-84-1, Sigma, St. Louis, MO, USA), and unspecific binding was blocked in blocking buffer (normal goat serum, Cat: PK-6101 Vectastain ^®^ Elite ABC + HRP, Vector Laboratories, Newark, CA, USA) for 1 h at RT. CDX sections were then incubated with primary antibody against Ki-67 (Cat: ab16667, Abcam) prior to incubating in anti-rabbit biotinylated secondary antibody. A rabbit-specific ABC + HRP Detection Kit (Cat: PK-6101 Rabbit IgG, Vectastain ^®^ Elite ABC + HRP, Vector Laboratories) was used, and Mayer’s hematoxylin was used as a counterstain (Hematoxylin Solution, Mayer’s modified, Cat: ab220365, Abcam). Diaminobenzidine substrate [DAB (3,3′-diaminobenzidine) Cat: SK-4100, Vector laboratories] was used as a stain to detect IHC reactivity. Stained tissue sections were scanned using an Aperio ScanScope AT Slide Imager (Leica Biosystems; Buffalo Grove, IL, USA) at 20× magnification. The resulting images were then viewed using Aperio ImageScope (v12.3.2).

### 2.14. Statistical Analysis

Statistical analyses were performed using GraphPad Prism software (v.10.2.3). The specific statistical tests are indicated in the figure legends with a *p* < 0.05 indicative of statistical significance.

## 3. Results

### 3.1. Identification of Novel Oxindole Ku-DBis

Given the cellular uptake limitations observed for Ku-DBis assessment in cellular assays and models [6] and building on our previous analysis of structure–activity relationships (SAR) [5], we focused on the synthesis and evaluation of new Ku-DBi derivatives, pursuing optimization of cellular uptake to improve their drug-like properties. In our previous studies [5], we identified that the substitution of the methyl group of a pyrazolone ring with a bioisosteric trifluoromethyl group resulted in an increase in the potency against Ku–DNA binding and DNA-PK activation. Therefore, we expanded our SAR studies, focusing primarily on the methyl–pyrazolone scaffold, and identified the oxindole scaffold highlighted in red (Figure 1A) for novel Ku-DBis.

Initial analysis of the oxindole modification alone demonstrated no loss of Ku inhibitory activity. Further, we replaced the fluoro group of phenyl Ring A of compounds **3618** and **3649** with a methoxy group to generate **3392** and **3395**, respectively (Figure 1A). Compounds **3618** and **3392** have a trifluoromethoxy group in place of the methoxy group on phenyl Ring B. The synthetic scheme of these compounds is depicted in Appendix A.

The impact of these modifications on Ku–DNA binding and DNA-PK inhibitory activity was assessed. The Ku–DNA binding inhibitory activity assessed by EMSA analysis (Figure 1B) demonstrated that all four compounds are capable of blocking Ku–DNA interactions. The replacement of the fluoro group of phenyl Ring A with a methoxy group in compounds containing methoxy substituents on phenyl Ring B resulted in a 50% increase in Ku inhibitory activity while the same changes had no effect on compounds with the trifluoromethoxy modified Ring B. Thus **3395** was identified as the most potent based on EMSA analyses of Ku–DNA binding inhibition (Figure 1C) with an IC_50_ value of 2.4 µM. The analysis of the DNA-PK inhibitory activity shows that the methoxy-derivatives (**3392** and **3395)** were ~2.5-fold more active compared to their fluoro-modified counterparts (**3618** and **3649**) (Figure 1D, Table 1). This was largely independent of Ring B modification with both **3392** and **3395** yielding IC_50_ values of 0.77 μM.

While the relative potencies of the individual compounds were similar, we observed a significant variation in the cellular uptake in the NSCLC H460 cell line as a function of the various substitutions on the phenyl Rings A and B in the novel oxindole derivatives (Figure 1E). Ku-DBis **3392** and **3618** containing the trifluoromethoxy group on phenyl Ring B crossed the cellular membrane and were retained in cells ~3-fold more efficiently than the methoxy-derivatives **3395** and **3649**. The oxindole scaffold with the substitution on phenyl Ring B thus appears to be an important determinant of cellular uptake. In each case, the replacement of the methoxy group of the phenyl Ring B with a trifluoromethoxy group increased the cellular uptake by ~3-fold (Figure 1E). Importantly, the uptake experiments were all performed in standard growth media supplemented with 10% FBS. In our previous studies of predecessor compounds [5], nearly undetectable uptake was observed in the presence of serum, and the drug treatments for all cellular experiments were conducted in serum-free media. A time course of uptake was also determined, and a difference was demonstrated in both the initial rates and the steady-state uptake, again with **3392** achieving higher cellular levels than **3395** (Figure 1F). Similar data were also obtained in the NSCLC cell line A549 (Appendix A). The prediction analysis of total polar surface area (TPSA) and CLogP using QuikProp (v5.1, Schrödinger Suite 2024) indicated increased lipophilicity for 3392, potentially enhancing cellular uptake and potency (Appendix A). Overall, these data indicate that **3392** possesses the best combination of potency and cellular uptake, and thus 3392 was selected for further biophysical and biological analysis.

### 3.2. Quantitative Biophysical Analyses of Novel Oxindole Ku-DBis

Quantitative biophysical analyses of the interaction between Ku and **3392** were assessed by Isothermal Titration Calorimetry (ITC) and Microscale Thermophoresis (MST) [12], and the results are presented in Figure 2A. The fraction bound as a function of Ku-DBi concentration was plotted, and a K_D_ of 2.1 *±* 0.3 µM was calculated, consistent with the IC50 values calculated from the EMSA analysis. We assessed the ligand-induced fluorescence-change specificity using the SDS-denaturation test (SD-test) (Appendix A). Since the ligand-induced fluorescence changes exceeded ± 20% of the average, we conducted the SD-test to assess their specificity. This helps to distinguish changes from direct protein-ligand interactions versus non-specific effects like protein loss from aggregation, adsorption, or spectral interference. The signal is attributed to specific binding when the fluorescence intensities of the target and complex match after denaturation.

We also assessed the Ku-**3392** interaction by nanoDSF to determine thermal stability and aggregation. The data presented in Figure 2B demonstrate that 5 µM of **3392** induced a shift in thermal stability, and higher concentrations, up to 25 µM, did not induce additional changes. These data reveal that the Ku-DBi **3392** directly interacted with Ku and are consistent with our previous data, obtained with our earlier Ku-DBi **245**, where a gel-based thermal shift assay was used to conclude direct Ku-DBi–Ku interaction [5].

### 3.3. Cellular Effect of Ku Inhibition—Single-Agent Cellular Activity across NSCLC Cell Lines

The improved uptake afforded by the oxindole derivatives used to treat the cultured cells in complete media allows for analysis of the impact of Ku inhibition across cell lines with varied genetic alterations. Our previous studies assessed the impact of single-agent Ku-DBis during short-term treatment in serum-free media and demonstrated limited single-agent anti-cancer activity independent of genetic alterations, even in genetically susceptible cell lines [6]. While we were able to demonstrate the ability of the Ku-DBis to sensitize the cells to IR and bleomycin using that experimental approach, the limited single-agent activity was potentially the result of insufficient target coverage as a function of the limited cellular uptake, a phenomenon that did not limit the oxindole Ku-DBi **3392**. The optimized Ku-DBi **3392** displayed similar activities in the p53-null H1299 NSCLC cell line, which remained highly resistant to the Ku-DBi treatment, and the A549 adenocarcinoma cells, which remained only slightly sensitive. The p53-null H1299 NSCLC and A549 adenocarcinoma cell lines displayed cell viabilities of ~90% and 50%, respectively, at the highest concentrations assessed (Figure 3A,B, Table 2). Interestingly, the H460 large cell lung carcinoma and ATM-null H23 cell lines were considerably more sensitive to the Ku inhibition by **3392** (Figure 3C,D) with IC_50_ values of ~8 μM (Table 2), a phenomenon that cannot be attributed to induction of direct DNA damage (Appendix A). The comparison of **3392** to the active-site DNA-PKcs kinase inhibitor NU-7441 showed some similarities and differences. The ATM-null H23 cells were ~2-fold more sensitive to NU-7441 than the other NSCLC lines, which all displayed IC_50_ values of ~4.5 (Table 2). The Ku-DBi resistance observed in the A549 and H1299 cell lines suggests that different genetic factors impact sensitivity, which is consistent with earlier research demonstrating a DNA-PKcs-independent function of Ku [13]. While H23 cells are ATM null, other genetic alterations could impact sensitivity to Ku-DBi treatment as the ATM wt H460 cells were equally sensitive. Similarly, the resistance observed in the p53 null H1299 cells could be impacted by other factors, as the p53 wt A549 cells were similarly resistant to the Ku-DBi treatment.

### 3.4. Combination Studies in NSCLC Cells

In light of the increased cellular uptake and single-agent activity observed in the novel oxindole-based Ku-DBis, we pursued therapeutic combinations with bleomycin and IR. The results demonstrate that pre-treatment with Ku-DBi **3392** increased the sensitivity to both bleomycin and IR. Initial experiments were conducted to confirm on-target cellular activity using Ku-null MEFs [14] as we previously described for the predecessor compounds [5]. The Ku-DBi-dependent increases in the sensitivity to bleomycin were only observed in Ku wild-type cells, and while the Ku80-null MEF were initially sensitive to bleomycin, the addition of Ku-DBi did not increase sensitivity (Appendix A). The mechanism of increased sensitivity was assessed in the H460 NSCLC cell line that was sensitive to both bleomycin and **3392**. The data presented in Figure 4A, B demonstrate an additive interaction as assessed by the highest single-agent (HSA) and Bliss additivity models [15]. Analyses combined with IR were conducted in the sensitive A549 NSCLC and ATM-null H23 NSCLC cells. While the A549 cells were more resistant to **3392** alone, these cells showed increased sensitivity when combined with a 2 Gy radiation dose (Figure 4C). Interestingly, in the ATM-null H23 cells, there was a minimal, statistically insignificant effect of the Ku-DBi treatment with 2 Gy, which was significantly sensitized with 5 Gy when combined with **3392** (Figure 4D).

Considering this increased sensitivity in the ATM-null cells, we assessed how genetic alterations in other DSB repair proteins impacted the Ku-DBi sensitivity. Analyses of the Ku-DBi sensitivity were performed in the BRCA1 mutant triple-negative breast cancer (TNBC) cell line MDA-MB-436 and the BRCA1 wild-type MDA-MB-468. Interestingly, the BRCA1 mutant cells were less sensitive than the BRCA1 wild-type cells to the single-agent Ku-DBi treatment and the active site DNA-PK inhibitor NU-7441 (Figure 5A,B, Table 2), while both cell lines were more sensitive to NU-7441 than to the Ku-DBis. The noticeable increase in the observed NU-7441 single-agent activity suggests an off-target effect for this inhibitor (Figure 5A,B). Considering these differences in cellular response, we assessed the impact of combination treatment with the radiomimetic agent, bleomycin. Again, clear differences were observed with a decrease in sensitivity to bleomycin mediated by Ku-DBi and NU-7441 in the BRCA1 mutant cells (Figure 5C), while an increase in sensitivity was observed in the BRCA1 wild-type cells (Figure 5D). To determine if these differences were also observed with a more relevant therapeutic agent, we assessed combination with doxorubicin (Dox). Dox does not directly induce DSBs, but DSBs result from repair of the Dox damage and intercalation [16]. Similar to the combination treatments with bleomycin, a decrease in Dox sensitivity was also observed in the mutant MDA-MB-436 cells (Appendix A). These data demonstrate that the genetic components of varying cells and tumor types contribute to significant differences in the single-agent and combination activities of Ku-DBis. While we highlight the BRCA1 status, there are other differences between these two breast cancer cell lines. While both are K-RAS wildtype and possess p53 alterations, MDA-MB-468 cells display high p53 expression of a DNA binding site mutant while MDA-MB-436 cells possess a frameshift mutation that significantly reduces p53 expression. Thus, analysis comparing non-isogenic cell lines makes definitive demonstration of the impact of any single mutation on drug sensitivity difficult. However, these data do effectively demonstrate that the Ku-DBi single-agent sensitivity is not tumor or cell-line agnostic, as observed with other DDR-targeted agents including ATR, CHk1, and WEE1.

### 3.5. Ku-DBi Impact on DDR and Cellular Proliferation In Vivo

The oxindole modifications that drove increased cellular uptake and activity led us to assess the ability to target Ku in vivo. The experimental design, which is depicted in Figure 6A, employed the A549 NSCLC adenocarcinoma xenograft model. Cells were implanted, and tumors were allowed to grow until they reached an average of ~600 mm^3^. Mice were then randomized to receive either vehicle or **3392** via direct intratumoral delivery. Following a 4 h incubation, the tumors in the mice from each group were either irradiated or mock irradiated. Two hours after irradiation or mock irradiation, the mice were sacrificed, and the tumors were excised and processed for IHC and Western blot analysis. The impact of the Ku-DBi treatment on DNA-PKcs autophosphorylation is a direct pharmacodynamic measurement of Ku inhibition [5]. The in vivo data demonstrated the expected increase in DNA-PKcs autophosphorylation events after irradiation. The pretreatment with **3392** effectively reversed the IR-induced DNA-PKcs autophosphorylation in the individual tumors (Figure 6B,C).

Analyses of the DNA damage marker **γ**-H2AX revealed the expected IR-induced increase that was further exacerbated in the combination-treated tumors (Figure 6D), consistent with our previously published data, where Ku inhibition increased ATM activation at the DSBs resulting in increased γ-H2AX staining [6]. Finally, to assess the impact on cellular proliferation, we assessed Ki-67 using both IHC and Western blot analysis (Figure 6E,F). There were limited effects with the single-agent 3392 or IR treatments, but the combination of both treatments resulted in a statistically significant decrease in the Ki-67 expression, as detected by IHC (Figure 6E) as well as by Western blot analysis (Figure 6F,G). These data provide the first in vivo evidence that targeting the Ku–DNA interaction increases IR sensitivity.

## 4. Discussion

Targeting the DDR to treat cancer has gained considerable traction following the successful deployment of PARP inhibitors (PARPi) in the treatment of breast, ovarian, and prostate cancers that harbor mutations in BRCA1 and 2. While the exact PARPi mechanism of action and the relevant DNA “signal” that PARPi exploits is debated [17,18,19,20], there is unequivocal data that these inactivating mutations render cells hyper-sensitive to PARPi [21]. As the DDR is potentiated with the activation of PI3KK, a wide array of small-molecular inhibitors targeting numerous DDR kinases have been reported, with DNA-PK being no exception [3]. The initial development of active site-binding ATP mimetics has resulted in high in vitro potency under assay conditions for screening and selection [22,23]. However, DNA-PK has a *K*_M_ for ATP of 25 µM, and intracellular ATP concentration can be as high as 5 mM [24], which makes ATP-competitive reagents less effective in cells and tissues despite impressive in vitro inhibitory constants. Consistently, recent reports from clinical trials assessing DNA-PKcs inhibitors in combination with IR and other chemotherapeutics have revealed inadequacy of target inactivation, with insufficient inactivation at tolerable doses [25,26,27], highlighting the limitations of ATP mimetics: specificity and selectivity. While in vitro IC_50s_ are often in the nM range, higher clinical doses are required to overcome ATP concentrations, which can result in high off-target activity and toxicity. It has been estimated that off-target activity can extend to dozens of kinases [28,29], which does not account for numerous other ATP-binding proteins that could be affected. Therefore, development of allosteric inhibitors represents one avenue to circumvent the ATP-competitive mechanism, and the size and complexity of the DNA-PKcs structure provides ample opportunity toward this end [30].

We have exploited a novel approach of targeting the requirement for Ku–DNA binding for DNA-PK activation and developed Ku-DBis capable of blocking the protein–DNA interaction [4,5,6,31]. While the development of protein–DNA interaction inhibitors has a long history of failure, many of these attempts were made against sequence-specific transcription factors (TF) [32]. Importantly, the Ku–DNA interaction is sequence independent and is dictated by DNA structure [33]. This allows for greater flexibility in chemical efforts to perturb the Ku–DNA interaction compared to a TF, where specificity for a short sequence of nucleotides is needed for interacting with the relatively small binding cleft. The Ku bridge-and-pillar structure is a collection of domains that come together to form a unique structure capable of encircling double-stranded DNA [33]. While other proteins possess the ability to thread onto DNA, including PCNA and hexameric DNA helicases, these proteins do not employ the same structural motifs to support DNA binding [34,35]. This unique aspect of the Ku–DNA interaction leads to the potential to identify unique chemical inhibitors that display high specificity and selectivity. Considering that DNA-PK is the only kinase in the human proteome that requires Ku bound to DNA for activation, this strategy portends the potential for high selectivity and specificity.

In this report, we provide evidence for the feasibility of developing small-molecule Ku inhibitors that possess enhanced cellular uptake and membrane permeability to support cellular target inactivation. We believe that the increased lipophilicity of 3392, as determined by total polar surface area (TPSA) and cLOGP determinations, contributes to the improved cellular uptake and potency (Appendix A). We demonstrate that Ku-DBis efficacy varied across different cell lines and tumor types that differ from the variations observed with DNA-PK direct active site inhibition, suggesting that DNA-PK-independent roles of Ku may define novel genetic predictors of sensitivity. Combinatorial activity was observed with a series of DSB-inducing treatments. The results spanned additive, antagonistic, and synergistic interactions, depending on the specific agent and cell line. These data suggest the potential for combination cancer treatment, but a detailed analysis of the predicting factors will be necessary to ensure optimal combinatorial activity. We also report the first demonstration of in vivo activity and on-target inactivation of DNA-PK by a Ku-DBi that reduced tumor cell proliferation in combination with IR (Figure 6). These data provide the impetus for further developments in medicinal chemistry efforts to optimize Ku-DBis for better systemic delivery and favorable pharmacokinetics.

## 5. Conclusions

The discovery of Ku–DNA binding inhibitors as a new and innovative strategy for DNA-PK inhibition has enabled us to explore an alternative approach for cellular sensitization to current cancer therapies, making Ku–DNA binding inhibitors potentially valuable for use in cancer therapeutics. The development of a new series of derivatives through chemical optimization resulted in enhanced cellular uptake with retained potent Ku-inhibitory activity and cellular response. This improvement allowed for better Ku-DBis assessment in cell culture conditions, where a potent single-agent activity in an ATM-deficient NSCLC cell line was observed, and a differential interaction of the Ku-DBi with DSB-inducing agents as a function of BRCA1 status was discovered. This suggests that different genetic backgrounds may impact combination therapy potential in diverse patient populations.

Finally, the optimization of Ku-DBis enabled evaluation in live organisms, supporting the potential application of chemical Ku–DNA binding inhibition to disrupt DNA-PKcs autophosphorylation and to modulate the DDR. This ultimately led to a reduction in tumor cell proliferation when used in combination with ionizing radiation (IR).

This study is the first to show in vivo activity and on-target inactivation mediated by a Ku-DBi and provides the foundation for further medicinal chemistry development of Ku-DBis as a potential anticancer therapeutic strategy in combination with radiotherapy and other DNA-damaging agents.

## Figures and Tables

**Figure 1 cancers-16-03286-f001:**
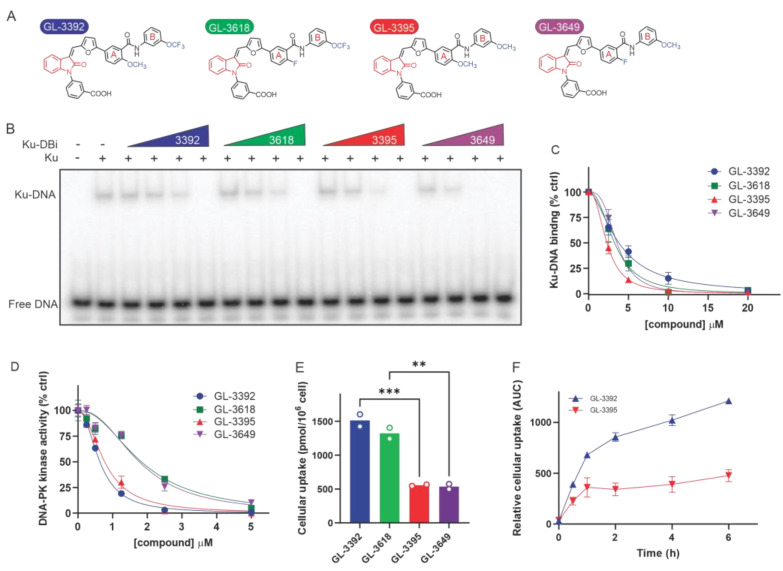
Novel chemical modifications of Ku-DBis improve cellular uptake while retaining potent inhibitory activity. (**A**) Optimized Ku-DBis chemical structures. (**B**) In vitro Ku–DNA binding inhibition as determined by EMSA. (**C**) Quantification of Ku–DNA binding inhibition. (**D**) Ku-DBi inhibition of DNA-PK kinase activity. (**E**) Cellular uptake of Ku-DBis assessed in H460 NSCLC cells. (**F**) Ku-DBis uptake time course in H460 cells. Data are presented as the mean of duplicate determinations. *** *p* = 0.0009, ** *p* = 0.0019 as calculated by one-way ANOVA with Šídák’s multiple comparisons tests.

**Figure 2 cancers-16-03286-f002:**
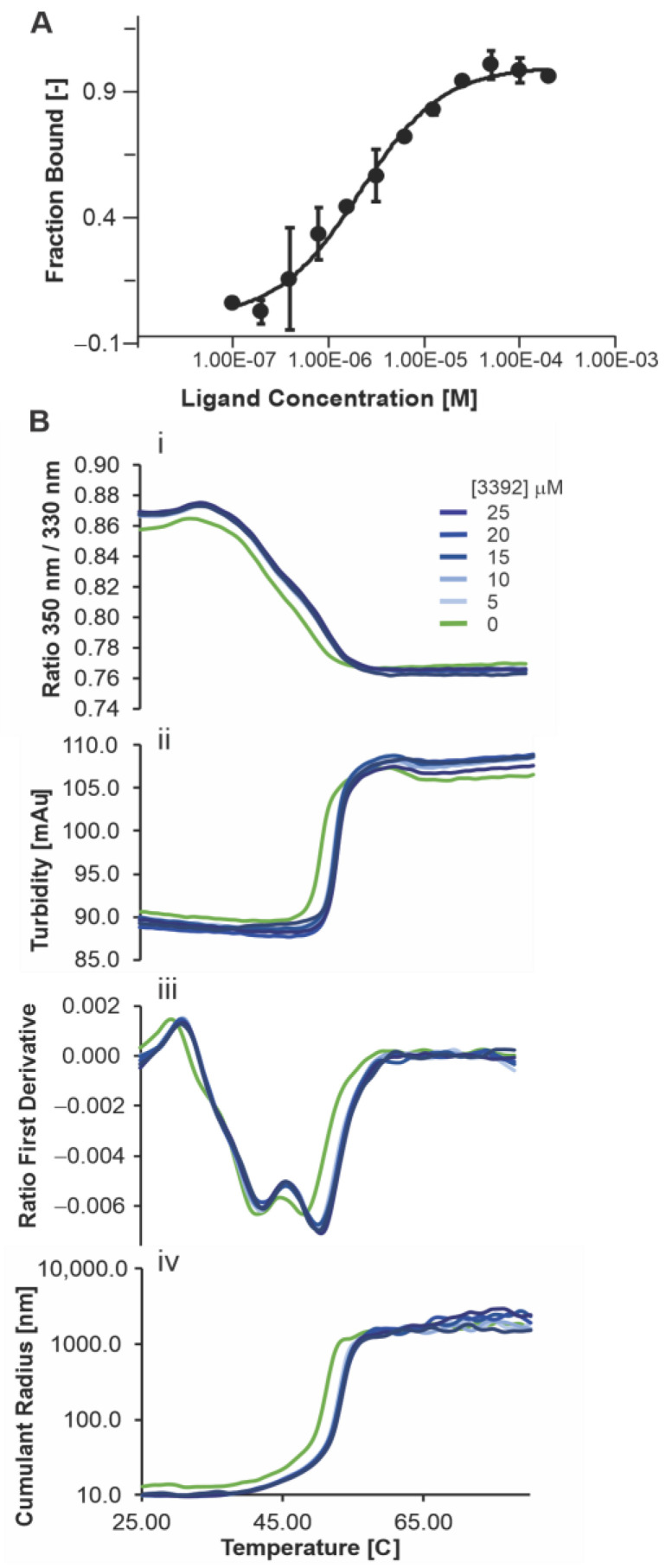
Biophysical analyses of the Ku-DBi 3392 interaction with Ku70–Ku80. (**A**) Interaction between Ku70–Ku80 with **3392** measured by MST (Microscale Thermophoresis) with a Kd of 2.1 ± 0.3 µM. The estimated bound fraction is plotted as a function of ligand concentration. (**B**) Impact of ligand **3392** on the thermostability and aggregation propensity of Ku70–Ku80: (**i**) ratio of intrinsic fluorescence at 350 nm divided by 330 nm, (**ii**) turbidity measurement, (**iii**) first derivative of the ratio, (**iv**) cumulant radius measured by DLS as a function of temperature.

**Figure 3 cancers-16-03286-f003:**
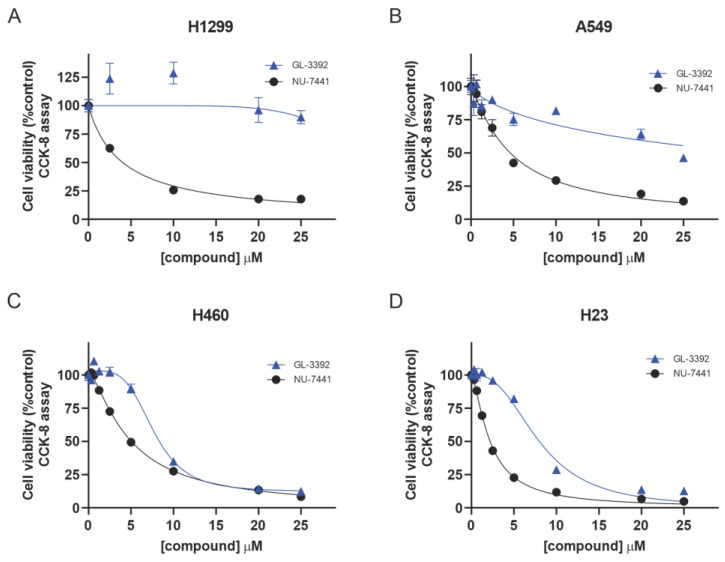
Single-agent Ku-DBi activity of 3392 in comparison with NU-7441 in NSCLC cells. The indicated cell lines (**A**) H1299, (**B**) A549, (**C**) H460 and (**D**) H23 cells were plated and treated with the indicated agent for 48 h, and cell viability was determined using CCK-8 assays as described in Materials and Methods. Data represent the mean ± SEM of triplicate determinations.

**Figure 4 cancers-16-03286-f004:**
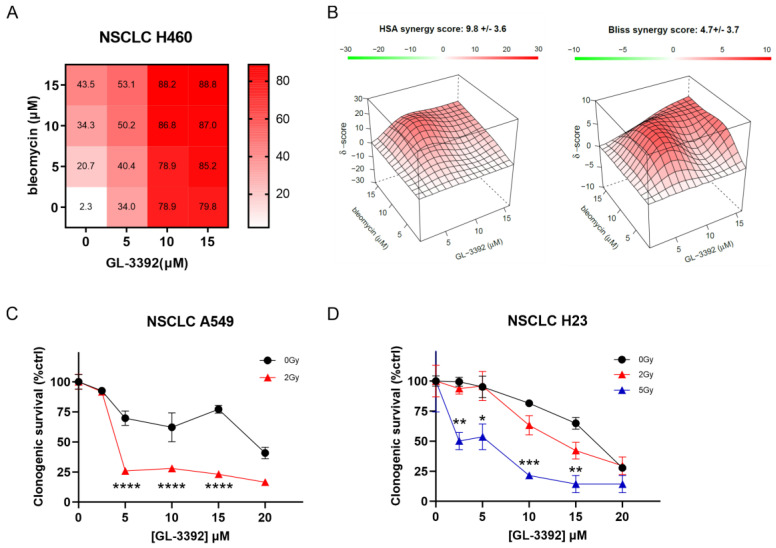
Ku–DNA binding inhibition enhances cellular sensitivity to IR in NSCLC models. (**A**) Isobologram analysis for **3392** and bleomycin combination in H460 cells and response matrix for inhibition. (**B**) Surface plot for HSA and Bliss independence–additive models for synergy assessment. Data were analyzed using the Synergy finder tool v3.0 (https://synergyfinder.fimm.fi, accessed on June–July 2024). (**C**,**D**) Analyses of **3392** in combination with IR in (**C**) A549 and (**D**) H23 cells. Tukey’s multiple comparisons test, simple effects within rows. **** *p*< 0.0001, *** *p*< 0.001, ** *p*< 0.005, * *p*< 0.05.

**Figure 5 cancers-16-03286-f005:**
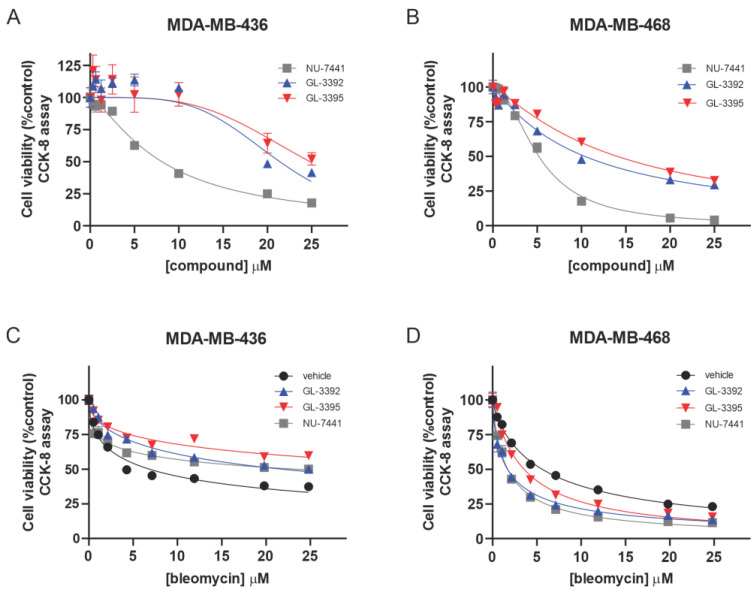
Single-agent and combination cytotoxic activities of Ku-DBi 3392 in comparison with NU-7441 in TNBC cell lines. (**A**,**B**) The indicated cell lines were plated and treated with increasing concentrations of the indicated single agents, and cellular viability was determined by CCK-8 assay. (**C**,**D**) Bleomycin-sensitization activity was determined for the Ku-DBi and NU-7441 in the indicated cell lines. Cells were plated and treated as described in Materials and Methods and processed as described above. Data are presented as the mean ± SEM of triplicate determinations.

**Figure 6 cancers-16-03286-f006:**
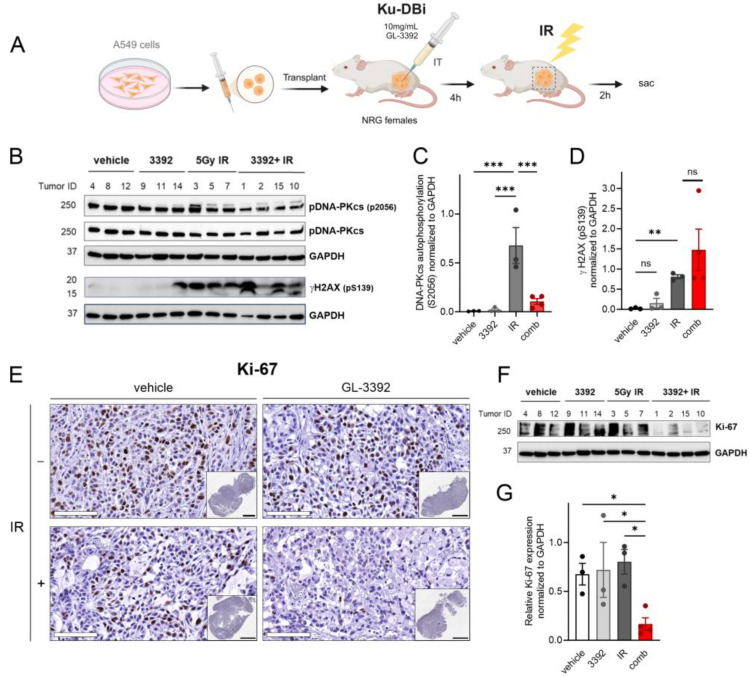
Impact of Ku-DBi in combination with IR on cell proliferation and early DDR events in NSCLC A549 CDX models. (**A**) Experimental design for combination Ku-DBi and IR treatment of NSCLC in vivo. (**B**) Western blot analysis from CDX tumor extracts assessing DNA-PKcs and **γ**-H2AX. (**C**,**D**) Quantification of the protein expression data presented in Panel B. (**E**) Representative 20X images of Ki-67 images, Sb: 100 mm. The insets show whole tumor sections for each treatment, Sb: 3 mm. Images were acquired using the Aperio ScanScope CS system. (**F**) Western blot detection of Ki-67 from the tumor tissue extracts. (**G**) Quantification of the data presented in panel F. Data are shown as mean ± SEM (vehicle n = 3, 3392 n = 3, IR n = 3, combination n = 4). Statistical analysis was performed using Fisher’s least significant difference test of the individual comparisons. Significant differences are indicated by * *p* < 0.05, ** *p* < 0.01, *** *p* < 0.001; ns: not significant. The original Western blot membranes can be found in Appendix A.

**Table 1 cancers-16-03286-t001:** Biochemical analysis of Ku-DBis.

Ku-DBi	Ku _70/80_ IC_50_ (µM) ^a^	DNA-PK IC_50_ (µM) ^b^
**GL-3392**	4.15 ± 1.07	0.77 ± 0.19
**GL-3618**	3.36 ± 0.89	1.72 ± 0.15
**GL-3395**	2.41 ± 0.77	0.77 ± 0.24
**GL-3649**	3.55 ± 0.40	1.74 ± 0.61

^a^ DNA binding was measured by EMSA as described in Materials and Methods. ^b^ DNA-PK kinase activity was determined by measuring the DNA-dependent phosphate transfer to a synthetic p53 peptide substrate, using purified DNA-PK. Values represent the mean and SD of triplicate independent experiments.

**Table 2 cancers-16-03286-t002:** In vitro cytotoxic activity of Ku-DBi GL-3392 compared to NU-7441 in NSCLC and TNBC cell lines.

Cancer Type	Cell Line	Inhibitor	IC_50_ (µM) ^a^	Cell Viability at 25 µM (%ctrl)
NSCLC	H460	GL-3392	7.7 ± 0.6	12.3 ± 0.6
		NU-7441	4.8 ± 1.0	8.3 ± 0.1
	A549	GL-3392	>50	46.2 ± 3.8
		NU-7441	4.6 + 0.9	13.6 ± 1.4
	H23	GL-3392	8.0 ± 0.6	12.7 ± 0.7
		NU-7441	2.2 ± 0.1	4.8 ± 0.5
	H1299	GL-3392	>50	89.9 ± 10.1
		NU-7441	4.0 ± 0.9	17.8 ± 5.4
TNBC	MDA-468	GL-3392	10.4 ± 1.5	29.4 ± 1.3
		NU-7441	5.2 ± 0.4	4.1 ± 0.9
	MDA-436	GL-3392	22.2 ± 1.4	43.9 ± 2.6
		NU-7441	8.1 ± 0.9	17.8 ± 1.3

^a^ Cells were plated and treated with increasing Ku-DBi GL-3392 or NU-7441 concentrations for 72 h after which CCK-8 determined cellular viability. Data are presented as the mean ± SD of triplicate determinations.

## Data Availability

The original contributions presented in this study are included in the article/Appendix A; further inquiries can be directed to the corresponding authors.

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
