# Peer review of "Impact of Optimized Ku–DNA Binding Inhibitors on the Cellular and In Vivo DNA Damage Response"

_cancers, 2024, doi:10.3390/cancers16193286_

Round 1
Reviewer 1 Report
Comments and Suggestions for Authors
The manuscript was aimed to examine the activity of Ku-DNA binding inhibitors on the cellular and in vivo DNA damage responses (DDR). Overall, the manuscript is well-prepared and illustrates a novel data and possibilities to sensitise cancer to DNA-damaging therapies, including certain chemotherapeutic agents and ionising radiation, as well.
I have the following questions and recommendations regarding this manuscript.
1) Since the authors showed the cytotoxic activity of GL-3392 against H460 and ATM-null cancer cells (Figures 3 C and D, respectively), I’m just wondering whether this inhibitor was able to induce DNA damage when it was used alone?
2) xenograft studies were performed at 2h of post-radiation only. Despite this time-point is in a proper fit with aim of this study, it will be interesting to show the changes in tumor weight and size of the time-points will be extended for 1-2 weeks. Additionally, IHC-staining for apoptosis markers (e.g., caspase-3) will be also highly desirable.
3) western blotting for in vitro studies illustrating an efficacy of the combination of Ku-DNA inhibitor and chemotherapies is also recommended to show an increased apoptosis death observed for this particular experimental condition when compared with single treatment.
4) additive, but not synergistic effects for Ku-DNA inhibitor and bleomycin shown in figures 4A and B are a bit disappointing. This narrows the potential use of these inhibitors in future. This issue should be highlighted and discussed in more detail in Discussion.
Author Response
Comment 1. The manuscript was aimed to examine the activity of Ku-DNA binding inhibitors on the cellular and in vivo DNA damage responses (DDR). Overall, the manuscript is well-prepared and illustrates a novel data and possibilities to sensitize cancer to DNA-damaging therapies, including certain chemotherapeutic agents and ionizing radiation, as well.
I have the following questions and recommendations regarding this manuscript.
Since the authors showed the cytotoxic activity of GL-3392 against H460 and ATM-null cancer cells (Figures 3 C and D, respectively), I’m just wondering whether this inhibitor was able to induce DNA damage when it was used alone?
Response 1. We appreciate all your comments and suggestions. We have not observed any indication of direct DNA damage induced by 3392 or its predecessor compounds, even when doses were increased to levels where single agent cytotoxic activity was observed. This is consistent with 3392 blocking repair of exogenously induced DNA damage. While it is possible that endogenous DSBs could be generated and their repair affected, the level is not detectable in our assays. We have added a supplementary figure showing the impact of single-agent on γ-H2AX levels as a marker of DNA damage assessed through western blot immunodetection and immunofluorescence assays in A549 and H460 cells (Supplementary Figure A6, Page 10 line 364-365). The highest dose of 3392 in these experiments 20 µM, induced greater than 95% cell killing in the H460 cells with no appreciable detection of γ-H2AX.
Comment 2. Xenograft studies were performed at 2h post-radiation only. Despite this time-point is in a proper fit with aim of this study, it will be interesting to show the changes in tumor weight and size of the time-points will be extended for 1-2 weeks. Additionally, IHC-staining for apoptosis markers (e.g., caspase-3) will be also highly desirable.
Response 2. We agree completely, however, intra-tumoral delivery is not an optimal route for larger scale studies because of the inherent variability or delivery. We are currently optimizing systemic delivery methods and the requisite pharmacokinetic studies to determine the optimal timing and concentrations to achieve sufficient target inactivation. Once established, we will assess tumor size and weight in combination studies, but these studies fall outside of the scope of this paper and we believe the data demonstrating in vivo target inactivation and effect on DNA damage and cell proliferation is an important advance.
Comment 3. Western blotting for in vitro studies illustrating an efficacy of the combination of Ku-DNA inhibitor and chemotherapies is also recommended to show an increased apoptosis death observed for this particular experimental condition when compared with single treatment.
Response 3. We have conducted experiments to assess the mechanisms of cell death induced by single agent Ku-DBis and minimal evidence of apoptosis was observed. As the combination studies were conducted at doses of Ku-DBis with limited single agent activity, the cell death observed by the combination agent, IR, bleomycin or doxorubicin, is likely to be exacerbated by Ku-DBi treatment. While IR and doxorubicin can induce apoptosis, bleomycin treatment induces cell death through non-apoptotic pathways. This is consistent with our previously published work using Ku–DBI predecessor and effects of combination observed in Incucyte Live cell analysis and cellular viability assays (Mendoza-Munoz et al. NARC 2023). This is an important issue in the mechanism of Ku-DBi activity, that it can sensitize to DSB agents independent of the mechanism by which they induce cell death.
Comment 4. Additive, but not synergistic effects for Ku-DNA inhibitor and bleomycin shown in figures 4A and B are a bit disappointing. This narrows the potential use of these inhibitors in future. This issue should be highlighted and discussed in more detail in Discussion.
Response 4. Thanks for bringing this suggestion. We have modified the Result section of the manuscript and also highlight this important distinction in the Discussion section (Page 15, lines 512-516).
Reviewer 2 Report
Comments and Suggestions for Authors
The manuscript by Mendoza-Munoz and co-authors describes a new inhibitor of the double-strand The manuscript by Mendoza-Munoz and co-authors describes a new inhibitor of the double-strand break DNA repair complex - KU70/80-DNA-PK. One of the main characteristics of cancer cells is a deficiency in one or another DNA repair mechanism, which allows cancer cells to acquire new mutations and adapt to a stressful environment. This DNA repair deficiency makes cancer cells sensitive to DNA damaging agents, IR or chemotherapy. The strategy of inhibiting additional DNA repair pathways in cancer cells is quite attractive because multiple defects in the DNA repair system make cancer cells even more sensitive to anti-cancer therapy. The manuscript is important and well written. The authors have performed several experiments to comprehensively analyze the interaction and inhibitory potential of Ku-DBis.
The main concern is a conclusion on the effect of Ku-DBis in ATM-deficient and BRCA1-deficient cells. Although the cell lines selected by the authors indeed had a defect in these genes (ATM and BRCA1), the very important conclusions cannot be drawn only on one cell line without deleting these genes in the era of simple CRISPR-Cas9 gene editing technology. There are several differences in the genome of the selected cell lines that could explain the observed differences in resistance to Ku-DBis. The authors should generate cell lines with genetic mutations of ATM and BRCA1 or study more cancer cell lines with similar mutations, or remove such preliminary conclusions.
Author Response
Comment 1. The manuscript by Mendoza-Munoz and co-authors describes a new inhibitor of the double-strand break DNA repair complex - KU70/80-DNA-PK. One of the main characteristics of cancer cells is a deficiency in one or another DNA repair mechanism, which allows cancer cells to acquire new mutations and adapt to a stressful environment. This DNA repair deficiency makes cancer cells sensitive to DNA damaging agents, IR or chemotherapy. The strategy of inhibiting additional DNA repair pathways in cancer cells is quite attractive because multiple defects in the DNA repair system make cancer cells even more sensitive to anti-cancer therapy. The manuscript is important and well written. The authors have performed several experiments to comprehensively analyze the interaction and inhibitory potential of Ku-DBis.
The main concern is a conclusion on the effect of Ku-DBis in ATM-deficient and BRCA1-deficient cells. Although the cell lines selected by the authors indeed had a defect in these genes (ATM and BRCA1), the very important conclusions cannot be drawn only on one cell line without deleting these genes in the era of simple CRISPR-Cas9 gene editing technology. There are several differences in the genome of the selected cell lines that could explain the observed differences in resistance to Ku-DBis. The authors should generate cell lines with genetic mutations of ATM and BRCA1 or study more cancer cell lines with similar mutations, or remove such preliminary conclusions.
Response 1. Thanks for all your comments and suggestions. We agree and have modified our presentation of the data and conclusions drawn from these experiments more accurately reflect the limitations of non-isogenic cell lines. We also have highlighted the other know differences in the cell lines as to no bias our presentation. Incorporated changes in the manuscript can be found on page 10-11, lines 369-375, and page 12-13, lines 430-440.
Reviewer 3 Report
Comments and Suggestions for Authors
The manuscript titled “ Impact of optimized Ku-DNA binding inhibitors on the cellu-2 lar and in vivo DNA damage response” reads well but there are aa few major comments that should be addressed. Overall, while the results and discussion sections provide valuable insights into the development and evaluation of novel Ku-DBis, addressing the below mentioned flaws would strengthen the scientific rigor and clarity of the paper.
Major comments:
1. There is inconsistency in the cell line names (e.g., "H460" vs. "NCI-H460"). It's good practice to use consistent terminology to avoid confusion.
2. In Section 2.5, the authors mention that the cells were plated and incubated for 24 hours before treatment, but there is no mention of the cell density used for plating. To correctly reproduce the experiment, cell density is important as it can affect cell behavior and treatment response.
3. The control groups are not described in detail in the cell viability and irradiation studies. It is important to specify what conditions the control groups were exposed to, including any vehicle treatments, to ensure that the observed effects are due to the experimental treatment.
4. For in vivo studies, the number of animals used per group and how they were randomized should be specified for assessing the robustness and ethical considerations of the study.
5. In section 3.1, the results section often describes trends and observations, such as the relative potency of different compounds, without providing sufficient quantitative data in the sentence to back these results. For example, statements like "3395 being the most potent" or "3392 possesses the best combination of potency and cellular uptake" are not accompanied by IC50 values or fold differences. Similarly, phrases like "no loss of Ku inhibitory activity" or "a modest effect on Ku inhibitory activity" are not quantified, making it difficult to gauge the actual impact of the modifications. Providing specific percentage changes or comparative inhibition constants would make the findings clearer and more interpretable.
6. The results and discussion sections heavily reference previous studies without integrating the new findings fully. While it's important to relate new data to past research, the current findings should be interpreted in their own right and then discussed in the context of existing literature. This will help to highlight the novelty and impact of the current study more clearly.
7. While the results describe the effects of different compounds and modifications, there is limited discussion on the underlying mechanisms driving these effects. For example, why certain modifications lead to better cellular uptake or potency remains speculative. Including more mechanistic insights, possibly supported by molecular modeling or additional biochemical assays, would enhance the understanding of the results.
8. The authors might be potentially biased in presenting the data. The discussion is selectively highlighting the positive results, such as the effectiveness of the novel compounds, without equally addressing limitations or negative findings. For a balanced discussion, both strengths and limitations should be acknowledged, and potential alternative explanations for the findings should be considered.
Author Response
The manuscript titled “ Impact of optimized Ku-DNA binding inhibitors on the cellu-2 lar and in vivo DNA damage response” reads well but there are aa few major comments that should be addressed. Overall, while the results and discussion sections provide valuable insights into the development and evaluation of novel Ku-DBis, addressing the below mentioned flaws would strengthen the scientific rigor and clarity of the paper.
Comments 1. Major comments: There is inconsistency in the cell line names (e.g., "H460" vs. "NCI-H460"). It's good practice to use consistent terminology to avoid confusion.
Response 1. Thank you for bringing this to our attention. We have included the suggested changes in the manuscript to make the terminology more consistent.
Comments 2. In Section 2.5, the authors mention that the cells were plated and incubated for 24 hours before treatment, but there is no mention of the cell density used for plating. To correctly reproduce the experiment, cell density is important as it can affect cell behavior and treatment response.
Response 2. We agree and apologize for the lack of detail. This information is included in the revised manuscript. Changes can be found on Page 4, Section 2.5, lines 151-152, Section 2.6, lines 158-160, Section 2.7, lines 173-174, Section 2.8, lines 181-182.
Comments 3. The control groups are not described in detail in the cell viability and irradiation studies. It is important to specify what conditions the control groups were exposed to, including any vehicle treatments, to ensure that the observed effects are due to the experimental treatment.
Response 3. An updated paragraph has been included in Materials and Methods section, page 5, Section 2.9 as indicated below:
“The in vivo studies were conducted as approved by the Institutional Animal Care and Use Committee at Indiana University School of Medicine. A549 cells (~2.5 × 106) in 50% Matrigel were injected into the hind flanks of 8–10-week-old NOD.Cg-Rag1tm1Mom Il2rgtm1Wjl/SzJ (NRG) mice as previously described [10]. Tumor volumes were monitored by electronic caliper measurement. Mice were randomized into groups of 3 or 4 using a random group generator; 6 different scenarios were generated, and the one with the most similar tumor volume averages across groups was selected. Treatments were initiated when tumors were ~600 mm3. Tumors were administered Ku-DBi (30 µL of 10 mM) or vehicle (30 µL of DMSO) via intratumoral injection 4 h before 5 Gy irradiation of the tumor with a single dose of X-rays (250 kVp; dose rate=1.4 Gy/min; 2 cm x 2 cm field) under isoflurane anesthesia. Mice that did not receive the irradiation treatment were placed under isoflurane anesthesia for the same length of time as the irradiated mice (~8 minutes). Mice were sacrificed 2 h post-irradiation (or 2 h post-sham irradiation), tumors were excised, and processed for immunohistochemistry (IHC) assays. Cell-free protein extracts were prepared to assess DNA-PKcs autophosphorylation and γ-H2AX levels.”
Comments 4. For in vivo studies, the number of animals used per group and how they were randomized should be specified for assessing the robustness and ethical considerations of the study.
Response 4. Mice were randomized into groups of 3 or 4 using a random group generator; 6 different scenarios were generated, and the one with the most similar tumor volume averages across groups was selected. Four mice were placed in the Ku-DBi group, and 3 each in the other 3 groups.
Comments 5. In section 3.1, the results section often describes trends and observations, such as the relative potency of different compounds, without providing sufficient quantitative data in the sentence to back these results. For example, statements like "3395 being the most potent" or "3392 possesses the best combination of potency and cellular uptake" are not accompanied by IC50 values or fold differences. Similarly, phrases like "no loss of Ku inhibitory activity" or "a modest effect on Ku inhibitory activity" are not quantified, making it difficult to gauge the actual impact of the modifications. Providing specific percentage changes or comparative inhibition constants would make the findings clearer and more interpretable.
Response 5. We have moved the table of IC50 data to the main manuscript from the Supplemental data. This table can be found on page 11 as Table 2. We also have included specific quantitative assessments of the differences between Ku-DBis based on the data presented. These changes can be found on Results section, Page 8, lines 304, 305, 307, 315, 319.
Comments 6. The results and discussion sections heavily reference previous studies without integrating the new findings fully. While it's important to relate new data to past research, the current findings should be interpreted in their own right and then discussed in the context of existing literature. This will help to highlight the novelty and impact of the current study more clearly.
Response 6. We appreciate this point and have modified our presentation to highlight the novelty of the data and the potential impact. in the Discussion section.
Comments 7. While the results describe the effects of different compounds and modifications, there is limited discussion on the underlying mechanisms driving these effects. For example, why certain modifications lead to better cellular uptake or potency remains speculative. Including more mechanistic insights, possibly supported by molecular modeling or additional biochemical assays, would enhance the understanding of the results.
Response 7. Thank you for the suggestion, we have added a mechanistic interpretation for the SAR data. Essentially, we believe that the increased lipophilicity of 3392, as determined by total polar surface area (TPSA) and cLOGP determinations contribute to the improved cellular uptake and potency. We have added this to the revised manuscript in the Discussion section.
Comments 8. The authors might be potentially biased in presenting the data. The discussion is selectively highlighting the positive results, such as the effectiveness of the novel compounds, without equally addressing limitations or negative findings. For a balanced discussion, both strengths and limitations should be acknowledged, and potential alternative explanations for the findings should be considered.
Response 8. We thank the review for this insight and have modified our presentation to present a more balanced view of the strengths and weaknesses of the data as well as alternative explanations.
Round 2
Reviewer 2 Report
Comments and Suggestions for Authors
After revision, the author made significant improvements to the manuscript and responded to the reviewers' questions.
Reviewer 3 Report
Comments and Suggestions for Authors
The manuscript can be accepted in its present form